# Table tennis coaching system based on a multimodal large language model with a table tennis knowledge base

**Wenlong Ma[1], Yang Liu[2], Qing Yi[3], Xutao Liu[4], Wei Xing[5], Rongji Zhao[6], Huan Liu[7], Rongzhi Li[1]** *

1 School of Physical Education, Shanghai University of Sport, Shanghai, China, 2 College of Physical Education and Sports, Beijing Normal University, Beijing, China, 3 Faculty of Sports and Exercise Science, Universiti Malaya, Kuala Lumpur, Malaysia, 4 School of Physical Education, Jiangsu University of Science and Technology, Zhenjiang, China, 5 College of Physical Education and Health Engineering, Taiyuan University of Technology, Taiyuan, China, 6 Department of Physical Education, Tongji University, Shanghai, China, 7 College of Physical Education and Health, Hubei Business College, Hubei, China

* lirongzhi@sus.edu.cn

**Data Availability Statement:** The de-identified raw data used in this study will be made available in a public online repository following publication, Preprocessing and analysis code has been made

## Abstract

Table tennis is one of the most popular sports in the world, and it plays a positive role in the overall development of people's physical and mental health. This study develops an AI table tennis coaching system using a Multimodal Large Language Model with a table tennis knowledge base, aiming to provide precise training guidance and match strategies for table tennis beginners. Method: By using visual recognition technology, motion capture technology, and advanced multimodal large language models with a comprehensive professional table tennis knowledge base, the system accurately identifies common errors made by beginners and provides targeted training guidance. Result: The AI Table Tennis Coaching System demonstrates high accuracy in identifying mistakes made by beginner players, particularly in recognizing arm-related errors and racket-related errors, with accuracies reaching 73% and 82% respectively. Conclusion: The system operates at low costs, is easy to deploy, and offers a high cost-performance ratio, providing effective technological support for table tennis teaching and training. The AI table tennis coaching system is expected to play a significant role in enhancing training efficiency, promoting athlete skill improvement, and popularizing the sport. Future research will focus on improving the accuracy of footwork recognition in AI table tennis coaching systems and expanding their capability to provide training guidance for high-level athletes, thereby promoting the overall advancement of table tennis.

## 1 Introduction

Table tennis is one of the most popular sports globally [1]. Regular practice of table tennis has a significant positive impact on the overall development of people's physical and mental health. From the perspective of physical health, practicing table tennis can improve hand-eye coordination, strengthen the muscles in the legs, arms, and core, and burn calories [2]. Table tennis

available in a public GitHub repository:https://github.com/mwlsus/ttcs_by_llm.

**Funding:** The author(s) received no specific funding for this work.

players show better cardiovascular endurance, muscle strength, and flexibility compared to physically peers not engaged in regular sports training [3]. Table tennis not only provides physical benefits but also offers numerous cognitive advantages, positively impacting brain function and overall mental health [4]. From a psychological perspective, table tennis training can significantly enhance concentration, allowing players to focus better during matches. Athletes who practice table tennis regularly also exhibit better mental resilience when facing difficulties and challenges. Interest in learning table tennis can inspire individuals to be more willing to participate in the sport, thereby demonstrating the numerous benefits of playing table tennis. An effective and scientifically-based table tennis training method can stimulate table tennis learners' interest. However, the current table tennis training methods are relatively limited, still adhering to a teacher-centered approach where instructors lecture while students listen. Additionally, teachers predominantly follow traditional textbook-based teaching methods, which lack personalized instruction and place students in a passive learning state, inevitably affecting their interest in learning table tennis skills [5]. With the continuous progress of society, many emerging scientific technologies are constantly evolving. The evolution of technology has markedly advanced table tennis training methods, integrating various technological tools to augment training efficiency and outcomes [6, 7]. The application of Virtual Reality (VR) technology enables learners to train within simulated environments [8], which not only heightens learner engagement but also allows for specialized skill practice unaffected by external conditions [9, 10]. However, VR technology exhibits significant limitations in delivering precise and personalized guidance. VR training environments primarily emphasize scene recreation and often lack the capability to dynamically adjust based on real-time feedback from learners. As a result, VR are unable to provide targeted recommendations for technical improvement. In recent years, AI has made significant advancements, it can provide personalized training recommendations and plans for beginners, strengthening their understanding of technical movement learning. Moreover, it lays a solid foundation for beginners to enhance their proficiency in technical skills. Artificial intelligence is the science of endowing programs with the ability to change themselves for the better as a result of their own experiences [11].

A significant breakthrough in the field of artificial intelligence, specifically in Large Language Models (LLMs), has showcased formidable capabilities in language comprehension and generation across various sectors [12]. The state-of-the-art multimodal large language models can not only understand and generate natural language but also process various data types such as images and text [13], bringing revolutionary changes to the intersection of sports and AI. In the field of sports, the combination of large language models with visual recognition and motion analysis technologies provides coaches with more precise and scientific data analysis tools [14, 15], thereby assisting athletes in personalized training and strategic planning. Furthermore, the use of multimodal LLMs extends beyond optimizing training and competition strategies [16]. They can also enhance competition analysis by generating comprehensive match reports and highlight reels through detailed video analysis, thereby improving spectator experiences and the quality of media coverage [17, 18]. Additionally, these technologies introduce innovative tools and methods for sports research, athlete health monitoring, and injury prevention, showcasing their extensive and impactful applications in sports [19].

Based on previous research, there is considerable study on motion capture technology and computer vision recognition for analyzing table tennis techniques. However, research applying advanced multimodal large language models to the field of table tennis is relatively scarce. Some studies have utilized multimodal data and neural networks to analyze specific table tennis techniques, such as the forehand stroke's length and position [20]. Currently, research that combines motion capture technology, computer vision recognition, and multimodal large language models in the context of table tennis is lacking, marking it as a relatively new area of

exploration. There is a scarcity of reports related to the training systems and datasets for table tennis, particularly for beginners. AI table tennis coaching system based on a Multimodal Large Language Model with a table tennis knowledge base is developed in this study. This system effectively identifies common errors among beginners and offers targeted guidance and training recommendations. By compiling and analyzing extensive data on table tennis techniques, this study has constructed a comprehensive dataset that supports further research and enhances the AI system's capabilities. Moreover, This study compares the performance of various LLMs in table tennis coaching, analyze the costs of applying multimodal LLMs in training. AI table tennis coaching system significantly improves training efficiency and quality while substantially reducing coaching costs. These advancements contribute to the popularization of table tennis and inject new vitality into the field of table tennis training.

This study explores the application of LLMs in table tennis training by integrating visual recognition technologies and motion analysis. This integration aims to provide precise training guidance and competition strategies for novice players. This research uses motion imaging technology to track the trajectory of the table tennis ball and employs open-source libraries such as OpenPose to capture athlete movements [21]. These technologies are combined into an AI table tennis coaching system, which is enriched with a multimodal large language model with a table tennis knowledge base.

## 2 Methods

Before conducting visual recognition and motion capture analysis of table tennis trajectories, a smartphone was positioned on a tripod to capture clear angles of beginners' forehand, backhand, and serving techniques. These videos served as the primary source of technical motion data for this study. Visual recognition technology was utilized for high-precision tracking of table tennis ball trajectories, combined with motion capture techniques such as OpenPose for real-time pose estimation of beginners. The multimodal large language model analyzed the data to identify common mistakes and provide personalized training recommendations. While using a single camera offers simplicity, it may reduce the accuracy in capturing certain movements, particularly footwork. Future improvements could involve the addition of multiple camera angles to enhance precision in detecting footwork and other complex body dynamics. For Figs 2 and 3,the individuals in this research have provided written informed consent (as outlined in the PLOS consent form) for the publication of these images. Below is a detailed description of the methodology:

### 2.1 The overall process of AI table tennis coaching system

The AI table tennis coaching system is a major innovation in this study. It is primarily constructed by three advanced technologies, including motion capture, computer vision trajectory recognition, and a multimodal large language model based on the table tennis knowledge base. The specific process is illustrated in Fig 1.

This section will provide a detailed description of how beginners can use the AI table tennis coaching system to identify and correct their technical errors. The system first gathers visual data, including images and diagrams, which depict the player's posture, racket position, and arm movements during a stroke. Then, this information is transformed into multimodal data and combined with pre-designed prompts for input into the large language model. The model assesses various types of errors by analyzing these inputs. The multimodal large language model at the core of this system is GPT-4 [22], chosen after comparative testing against a range of alternative models.

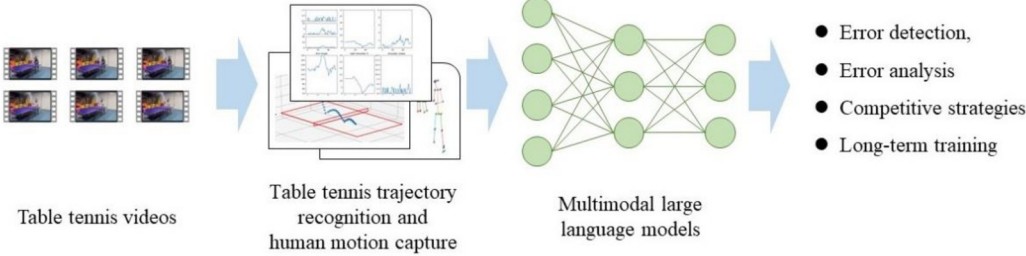

**Fig 1. Schematic diagram of the overall process of AI table tennis coaching.**

## 2.2 Visual recognition of table tennis trajectories

In the digitalization of table tennis training and competition, accurately recognizing the ball's trajectory is crucial for analyzing performance errors and providing targeted coaching [23]. A study has pointed out that the main challenge in predicting table tennis trajectories lies in measuring the ball's spin [24]. This study, however, focuses solely on the trajectory recognition of table tennis. The trajectory data will be utilized not only for error analysis but also as input for advanced language processing models, enhancing the focus and effectiveness of table tennis training. This study introduces a method for recognizing ball trajectory using a monocular camera combined with motion image processing, which aims to track the table tennis ball in real-time with high precision. While monocular camera-based methods might slightly underperform in reconstructing trajectories compared to multicamera systems, they offer significant advantages such as lower cost, easier operation, and the elimination of the need for complex setups or alignments [25].

Fig 2 illustrates the process used for visually recognizing the trajectories of table tennis balls. Initially, the method employs background subtraction to isolate the ball from its

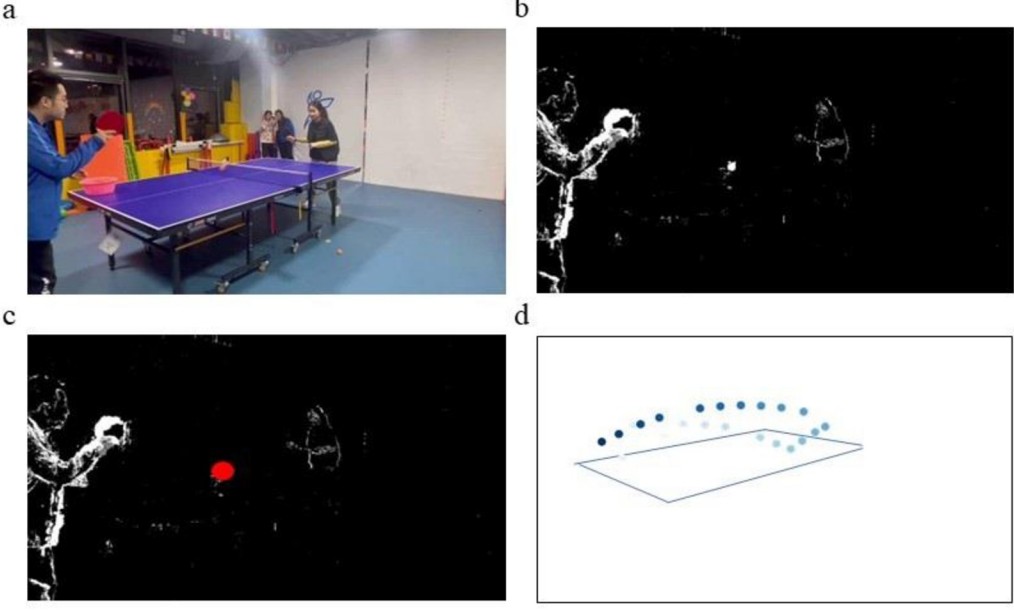

**Fig 2. Outlines the workflow of table tennis trajectory recognition.** A:original video. B:background subtraction and Gaussian blur. C:ball extraction. D:ball motion trajectory localization and recognition.

surroundings, mitigating environmental variations and enhancing system robustness. This technique differentiates consecutive frames, removing static backgrounds and retaining only the pixels of moving objects. As a result, it provides a clear foreground image, which aids in subsequent ball recognition. Additionally, applying Gaussian blur after background subtraction reduces image noise and smooths edges, improving the visibility of the ball's outline for more effective feature extraction and recognition.

During the ball positioning phase, the method uses dynamic difference information from adjacent frames to estimate the ball's location. It analyzes the morphological characteristics of areas with dynamic changes to preliminarily identify the ball's position. Further image processing techniques refine this positioning, enhancing the accuracy of the recognition. Finally, by integrating the ball's positional data across frames, the motion trajectory is accurately computed.

## 2.3 Motion capture and analysis

This section introduces the motion capture process based on the open-source library Open-Pose, which utilizes computer vision technology for real-time pose estimation of athletes. OpenPose can identify key points of individuals in videos, including the head, shoulders, elbows, wrists, knees, and ankles, thereby constructing a skeletal model of the athlete. While OpenPose provides detailed posture data, additional processing is often necessary to fully utilize this information. In this context, multimodal large language models, which excel in processing both image and text data, can be particularly useful. As illustrated in Fig 3, analyzing

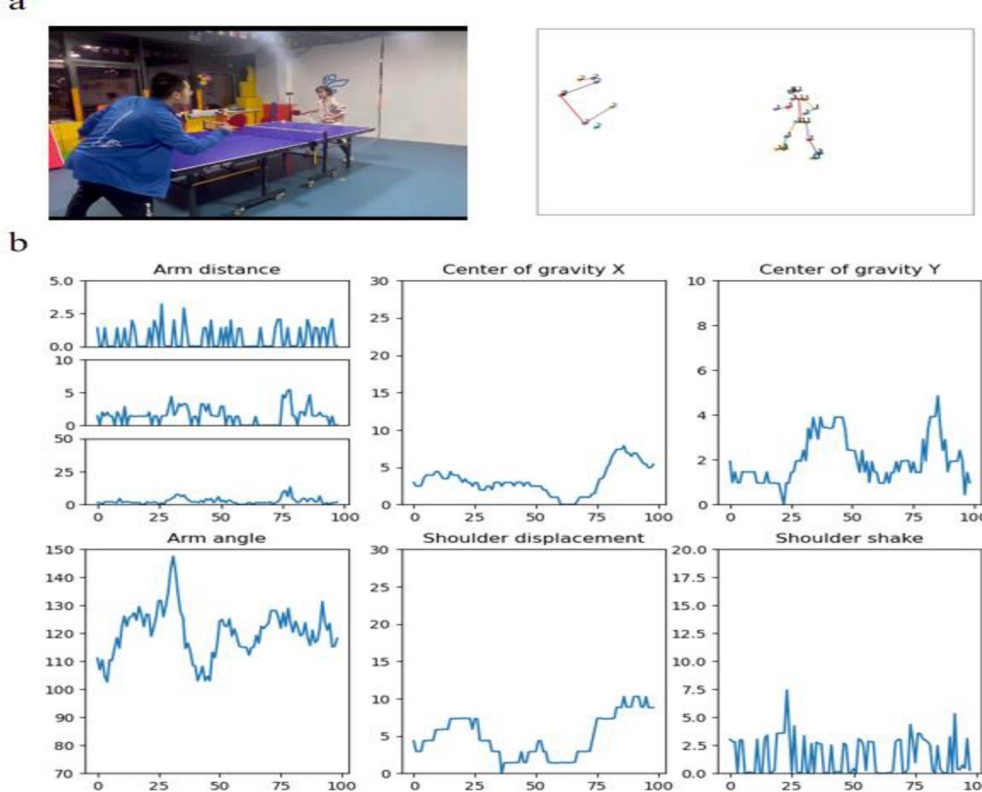

**Fig 3. The process of motion capture and analysis for table tennis.** A:OpenPose motion capture of a table tennis player during a stroke. B:motion charts derived from the capture, highlighting changes in shoulder and arm speeds, shifts in body center coordinates, arm angles, shoulder positioning, and body angle variations.

limb postures recognized by OpenPose in each frame enables us to extract key dynamic information. This includes the speed of arm and wrist movements, angles between arms, shifts in the body's center of gravity, and changes in body angles, all of which are plotted to provide insights into the athlete's performance. To develop an AI table tennis coach that is easily deployable for beginners, the requirements for video capture equipment have been minimized. Motion capture using a single 1080p, 30fps camera positioned diagonally across the table is sufficient.

The results from motion capture and analysis are integral to the AI-powered coaching system for table tennis, providing key data among other multimodal inputs such as player performance statistics and game tactics. This approach allows the system to efficiently identify and analyze unforced errors, even with minimal data samples. Consequently, it offers precise and personalized training recommendations based on its analyses.

## 2.4 Datasets

To develop and refine the AI table tennis coaching system, we created a detailed dataset centered on the frequent errors committed by novice players. This dataset, derived from video analyses, coach observations, and player feedback, classifies errors into three primary categories and eight specific unforced errors. The categories include:

- Technical Maneuver Errors: These encompass wrist, arm, racket orientation, and point of impact mistakes, focusing on proper racket handling and striking techniques.

- Timing and Rhythm Errors: These involve inaccuracies in striking timing and slow footwork, emphasizing the importance of choosing the right moments for striking and coordinating movements.

- Physical Condition Errors: Errors like slow body transitions and a high center of gravity are included here, underlining how physical fitness influences table tennis performance.

Each error type has been meticulously annotated and explained through video analysis by professional coaches, ensuring the dataset's quality and applicability. The process of data annotation was conducted by 10 experts qualified as table tennis referees. The proportion of each type of error in the overall dataset, as well as the Fleiss' Kappa values for each type of error, are presented in Table 1. It is evident that there is a high level of consistency among experts in error judgment. Due to the complexity of errors in table tennis, a single lost point may result from the combined influence of multiple error factors. This structured categorization not only enables the system to pinpoint learners' errors accurately but also facilitates the provision of customized improvement recommendations, thereby effectively boosting table tennis skills.

**Table 1. Corresponding multimodal input data for each error type.**

| Error Type | Error | Quantity | Proportion | Fleiss' Kappa |
|---|---|---|---|---|
| Technical Maneuver Errors | Wrist error | 6 | 9.68% | 0.86 |
| | Arm error | 30 | 48.39% | 0.82 |
| | Racket angle error | 38 | 61.29% | 0.91 |
| | Hitting position error | 17 | 27.42% | 0.79 |
| Timing and Rhythm Errors | Timing error | 17 | 27.42% | 0.79 |
| | Footwork lag error | 10 | 16.13% | 0.73 |
| Physical Condition Errors | Slow body transition error | 19 | 30.65% | 0.81 |
| | High center of gravity error | 7 | 11.29% | 0.81 |

## 2.5 Multimodal large language model with a table tennis knowledge base

The method of integrating a table tennis knowledge base into a multimodal large language model is particularly important for interpreting domain-specific knowledge questions. It effectively supplements the challenges that large language models may encounter when dealing with professional domain issues, such as insufficient evidence or generating inaccurate information (see Fig 4) [26]. Furthermore, multimodal input is another key aspect of AI table tennis coaching systems. By utilizing the results of table tennis trajectory recognition and player motion capture as inputs, the system obtains visual information and key action parameters of the player's movements. These multimodal inputs, combined with traditional textual information, provide LLMs with richer and more comprehensive data support. By integrating with multimodal learning and knowledge bases, LLMs can effectively operate in more complex scenarios. In a table tennis coaching system, an LLM based on a table tennis knowledge base can be used to understand various multimodal aspects such as action descriptions, game rules, technical movements, and subsequently generate training recommendations and match strategies based on this information. Due to the current limitations of large language models, our table tennis coaching system employs a knowledge base in the form of text-based question-and-answer descriptions. This encompasses various errors and performances observed during the offensive and defensive processes of novice table tennis players.

Trained on extensive textual data, LLMs like GPT-4 and Gemini have mastered complex linguistic structures and accumulated a broad range of knowledge. This expertise enables them to handle diverse tasks in understanding and generating language. Due to the extensive knowledge acquired by the GPT model during its training, including human motion and movement theories, our table tennis coaching model can effectively provide competent guidance in novice training. This is achieved by activating and integrating the pre-existing knowledge from the GPT model's pre-training through methods such as multimodal input, prompt engineering, and knowledge bases. The current multimodal large language models exhibit limited support for video content. Most video-level multimodal models rely on frame-by-frame recognition.

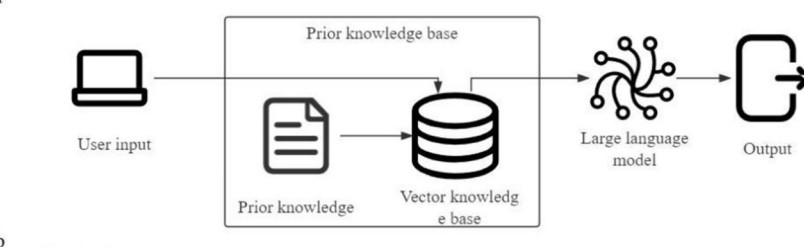

**Fig 4.** A: Schematic of a Large Language Model Based on a Vector Knowledge Base. B: Example of prompts for identifying arm related errors.

Therefore, our AI table tennis coaching system directly utilizes key frames and action summary diagrams as inputs for the multimodal large language model, aiming to achieve better outcomes. The key to building complex applications and addressing real-world challenges often involves using these models, here referred to as "agents." These agents, which operate based on prompts and keywords, are adept at understanding and generating natural language [27, 28]. This capability proves invaluable in various applications, ranging from responding to queries to creating content. Specifically designed agents find applications in areas such as simulations, virtual reality, and data analysis. For instance, in the AI-powered table tennis coaching system, agents analyze novice players' movements, provide training for matches, and compile career statistics by leveraging their extensive knowledge base. Agents are implemented through a series of pre-designed prompts, following a simplified CO-STAR framework, which includes Context, Outcome, Scale, Actor, and Resources. The Context provides essential information to help AI recognize situations in table tennis, including data and related image inputs and descriptions. The Outcome specifies the Agent's output, the Scale delineates the scope of the results, and the Actor guides the model to explain in the manner of a coach to the learner. Resources offer a corresponding knowledge base as background. Fig 4B presents an example of prompts used for identifying arm errors. The provided multimodal information, as shown in Table 2, includes multimodal image data such as immediate frames of missed shots and motion state charts obtained through OpenPose analysis, used to identify different types of errors. These image data are encoded in the universal base64 format for multimodal large models and, when combined with prompts and the knowledge base, are submitted in a JSON serialized form recognizable by the API to the LLM server for a response.

The workflow of the AI table tennis coaching system starts with software agents that analyze the ball's trajectory and the player's movements using visual recognition and motion capture technologies. These agents then use vector matching techniques—a method for retrieving relevant information from a database—to provide context for the LLM. This context facilitates the LLM's understanding and analysis of the gameplay. Subsequently, the LLM generates a detailed error analysis report on the player's performance and offers customized guidance and training suggestions. These recommendations are based not only on extensive professional knowledge but are also specifically tailored to the player's performance and technical characteristics, thereby enhancing the precision and personalization of the training support. By integrating large language models, knowledge bases, and multimodal inputs, the system delivers professional, personalized training recommendations and strategic advice, significantly improving both the efficiency and quality of training. Throughout the development of the AI table tennis coaching system, the GPT-4V model had no prior knowledge of any content within our dataset. Consequently, the large language model does not intentionally fit the training set.

Table 2. Corresponding multimodal input data for each error type.

| Error Type | Multimodal Input Data |
|---|---|
| Wrist error | Wrist speed-time graph, image of the hand at the moment of striking |
| Arm error | Arm speed-time graph, elbow angle-time graph, full-body image at the moment of striking |
| Racket angle error | Table tennis ball trajectory, image of the hand at the moment of striking |
| Hitting position error | Table tennis ball trajectory, image of the hand at the moment of striking |
| Timing error | Table tennis ball trajectory, image of the hand at the moment of striking |
| Footwork lag error | Center of gravity x-coordinate-time graph |
| Slow body transition error | Center of gravity x-coordinate-time graph, body tilt angle-time graph |
| High center of gravity error | Center of gravity x-coordinate-time graph, center of gravity y-coordinate-time graph |

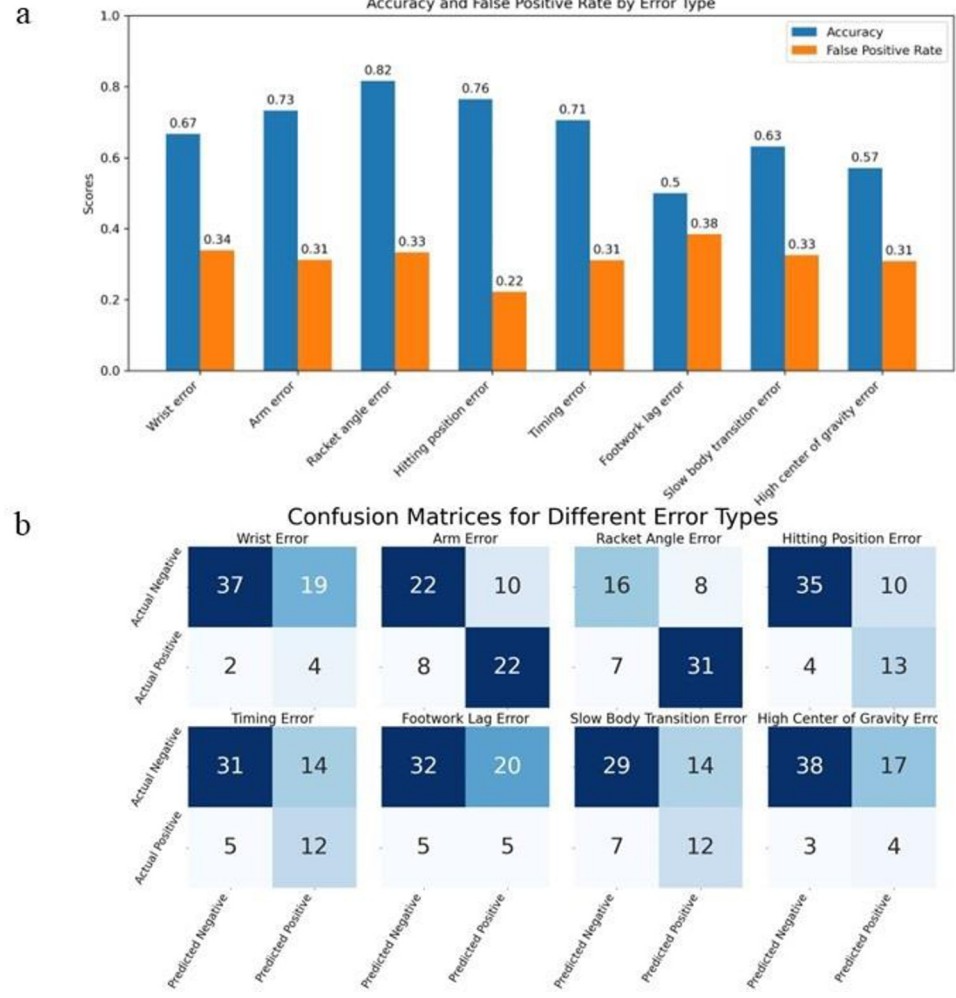

**Fig 5.** A:The accuracy and false positive rates of the AI table tennis coaching system in judging various types of errors in table tennis. B: Confusion matrix of AI table tennis coaching system for various errors.

# 3 Results

## 3.1 Identifying common errors in table tennis novices using AI table tennis coaching system

This section examines the accuracy and false positive rates of the AI table tennis coaching system in assessing various types of errors occurring in table tennis. All error datasets for beginner table tennis players were evaluated using the GPT-4 multimodal model, and the results are displayed in Fig 5A. The AI table tennis coaching system has a high overall prediction accuracy, with an average prediction rate close to 70%. This indicates that the use of a multimodal large language model is effective in identifying common mistakes made by beginner table tennis players. This study found that the system demonstrated high accuracy in identifying arm-related errors (such as improper arm speed and angle) and racket-related errors (such as incorrect racket face orientation) among novice table tennis players, with accuracies of 73% and 82%, respectively. These accuracy metrics are primarily designed for beginners and require further optimization for advanced players. Furthermore, in order to enhance the system's accuracy in identifying beginners, it is recommended to incorporate biomechanical data or utilize higher

**Table 3. Comparison of AI table tennis coaching system and sole use of GPT for error detection.**

| Model | Our Table Tennis Coach System | | GPT-4V | |
|---|---|---|---|---|
| Error Type | Accuracy Rate | Fleiss' Kappa | Accuracy Rate | Fleiss' Kappa |
| Wrist error | 0.667 | 0.86 | 0.333 | 0.43 |
| Arm error | 0.733 | 0.85 | 0.633 | 0.38 |
| Racket angle error | 0.816 | 0.88 | 0.526 | 0.44 |
| Hitting position error | 0.765 | 0.83 | 0.529 | 0.50 |
| Timing error | 0.706 | 0.75 | 0.353 | 0.31 |
| Footwork lag error | 0.5 | 0.74 | 0.4 | 0.29 |
| Slow body transition error | 0.632 | 0.73 | 0.368 | 0.37 |
| High center of gravity error | 0.571 | 0.71 | 0.286 | 0.33 |
| Average | 0.674 | 0.79 | 0.429 | 0.38 |

resolution tracking methods. Furthermore, Fig 5B presents the confusion matrix of our AI table tennis coaching system for various errors, facilitating a more standardized and visual representation of the system's performance in identifying table tennis errors. Overall, the identification of errors is meaningful; however, the system underperforms in recognizing footwork and gravity errors. This may be attributed to the occlusion of foot movements in some data samples.

As shown in Table 3, the overall average accuracy of the AI table tennis coaching system is 0.674. In contrast, the accuracy for error identification when directly utilizing GPT-4V without predefined prompts and a knowledge base is only 0.429. This indicates that the AI table tennis coaching system significantly enhances error recognition compared to the original GPT-4V multimodal model. This demonstrates that our AI table tennis coaching system possesses higher predictive accuracy and better reproducibility, whereas direct predictions using GPT-4 are inaccurate and exhibit lower Fleiss' Kappa and randomness.

### 3.2 Guidance for unforced errors in table tennis novices

Fig 6 illustrates a specific instance where a novice player loses a point, accompanied by the AI coach's analysis and tailored advice on the unforced errors made. For Fig 6, the individuals in this research have provided written informed consent (as outlined in the PLOS consent form) for the publication of these images.

To assess the accuracy and feasibility of these guidance strategies, we conducted an evaluation involving 10 table tennis experts. These experts reviewed the system-generated advice, focusing on its precision, practicality, personalization, and comprehensibility. Feedback from these table tennis experts has validated the efficiency and reliability of the AI table tennis coaching system in addressing unforced errors among novice players. This study highlights the potential benefits of integrating large language models with expert knowledge bases in sports training. Table 4 presents the details of the expert evaluations and the ANOVA analysis. Experts showed relatively higher approval for the accuracy and comprehensibility of the AI Table Tennis Coach. However, there remains room for improvement in the AI coach's personalization capabilities.

### 3.3 Personalized training recommendations based on comprehensive data analysis

In terms of short-term competitive strategies, the AI table tennis coaching system can accurately identify common error patterns among athletes during specific matches and promptly provide practical adjustment suggestions. For instance, in an analysis of an 11-point match,

Table 4. Details of expert evaluation of the AI table tennis coaching system.

| Evaluation Dimension | Evaluation Details | Score | Variance |
|---|---|---|---|
| Accuracy | The system demonstrates high precision in identifying and analyzing unforced errors. Additionally, the model's results are highly reproducible, ensuring consistency and reliability across evaluations. | 8.8/10 | 1.4 |
| Practicality | The recommended exercises and training plans are well-aligned with actual training needs, effectively assisting athletes in correcting errors and enhancing their competitive performance. | 7.6/10 | 2.5 |
| Personalization | The system provides tailored guidance based on each athlete's specific circumstances, addressing individual training requirements and optimizing personalized development. | 8.0/10 | 3.7 |
| Comprehensibility | The system's advice is clearly and concisely presented, making it easy to understand and significantly aiding novice players in improving their technical skills. | 8.6/10 | 3.9 |

the AI table tennis coaching system not only identifies the specific reasons for a player's lost points but also offers tactical adjustment plans based on the opponent's characteristics and the match situation. Such immediate feedback and targeted guidance help athletes quickly correct mistakes in subsequent matches, thereby improving their chances of winning. Fig 7 presents the AI table tennis coaching system's fully automated loss analysis, error cause analysis, and recommendations for a table tennis novice in an 11-point match. In Fig 7, it can be seen that in an 11-point match analyzed by the AI table tennis coaching system, a total of 41 points were lost. Among these, there were 9 errors each due to arm movements and racket angle, 7 errors each due to hitting position and slow body transitions, 3 errors due to mistiming of strokes, 1 error due to lagging footwork, 2 errors due to wrist movements, and 3 errors due to high center of gravity.

## 3.4 Performance evaluation of error recognition in multimodal large language models

We evaluated the error recognition performance of three advanced multimodal large language models—GPT-4, Gemini Pro [29], and Qwen-VL [30]—within an AI table tennis coaching

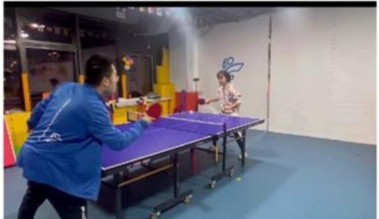

Advice from an AI Table Tennis Coach:

- Your shoulders are overly stiff, indicating that they are not relaxed enough, leading to a lack of fluidity in your strokes. The upper arm should lead the forearm in a coordinated effort, varying between primarily using the forearm or the upper arm depending on the situation, to ensure both coordination of movements and effective power transmission.
- During practice, attention should be paid to the angle and direction of the racket face, ensuring it is appropriately tilted forward when striking the ball to enhance both the power and control of your shots.
- For forehand strokes, the ball should be hit in the early stages of its ascent to utilize the rebounding force for an effective attack. By practicing early preparation and adjusting your stance, you can improve your anticipation and timing control of the ball.
- A high center of gravity can affect the stability and power of your shots. The correct approach is to keep your knees slightly bent and your body slightly leaning forward when hitting the ball, using a coordinated effort of your legs and waist to effectively transfer your center of gravity. Practicing center of gravity transfers without the ball can help improve control over your body's center of gravity.

Fig 6. AI table tennis coach analysis and guidance on a novice's point loss process.

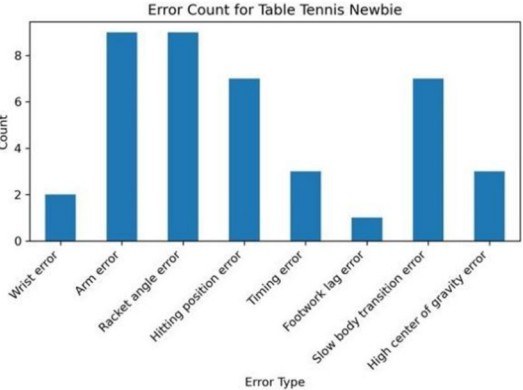

**Fig 7. Automated loss analysis, error cause analysis and recommendations for a table tennis novice in an 11-point match.**

system. Our analysis focused on two primary metrics: overall prediction accuracy and the false positive rate. To conduct this assessment, we selected a comprehensive random sample of video data showcasing various table tennis strokes. The aforementioned models were then employed to identify and classify errors in these strokes. As illustrated in Fig 8, GPT-4 outperforms in both key performance metrics. Specifically, GPT-4 achieved the highest prediction accuracy, reaching 70%, indicating its superior capability in accurately identifying errors in table tennis actions. Additionally, GPT-4 had the lowest false positive rate, at only 28%, meaning it is less likely to mistakenly classify normal actions as errors. These findings justify the selection of GPT-4 as the core model for the AI table tennis coaching system. Given its superior performance and cost-effectiveness, GPT-4 offers an economical and effective solution for integrating into the foundational model of the system.

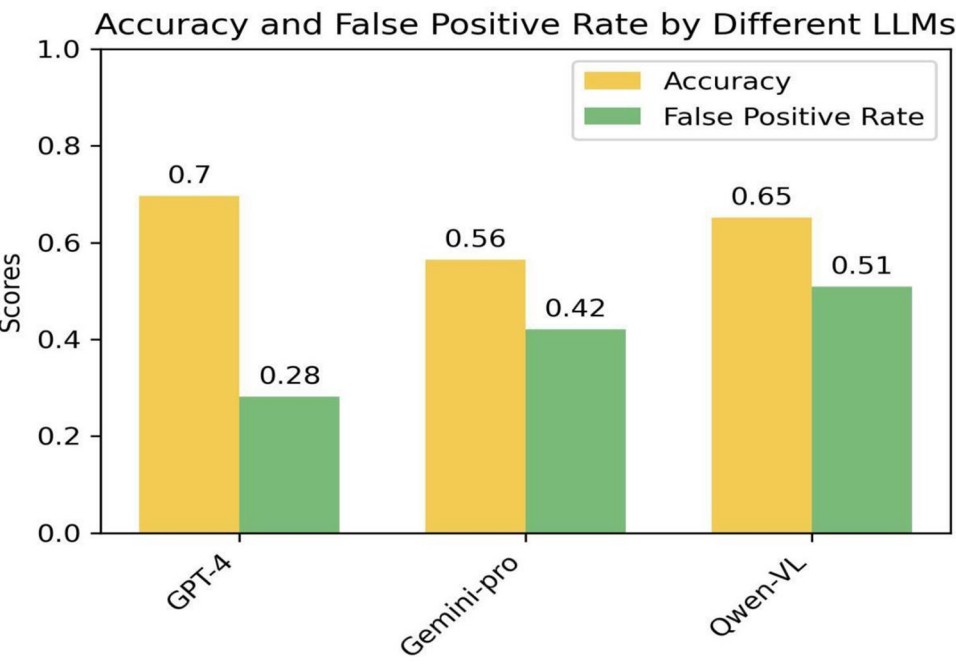

**Fig 8. Error recognition performance of different multimodal large language models.**

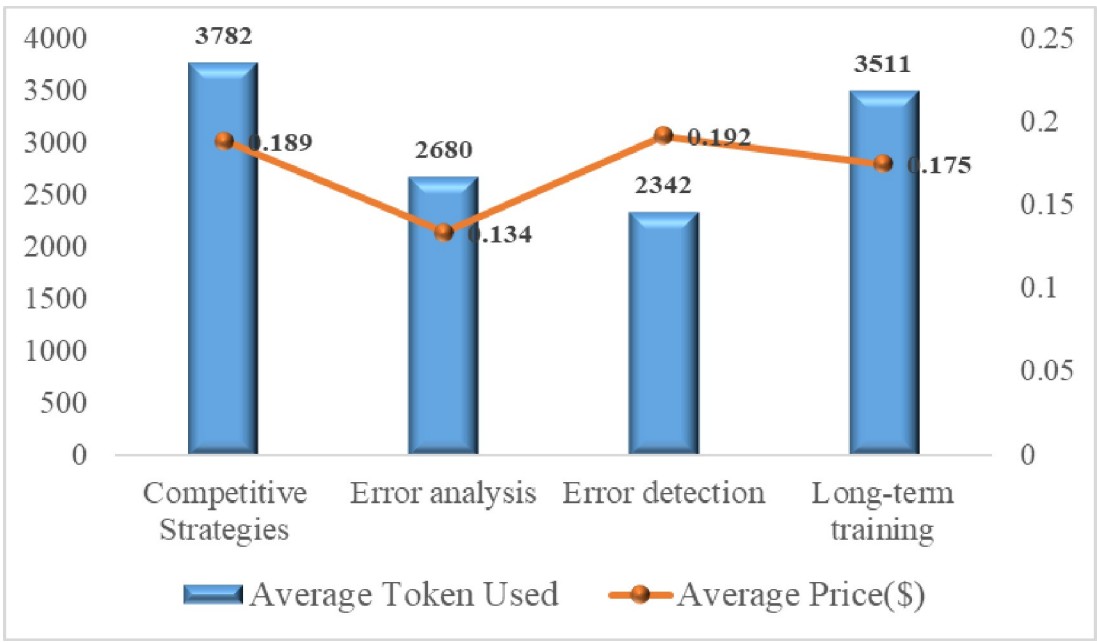

**Fig 9. Average token usage and costs in various processes of AI table tennis coaching system.**

## 3.5 Cost analysis of AI table tennis coaches

The AI table tennis coaching system utilizes advanced algorithms and data analytics capabilities to provide customized training programs and match strategies for beginners, helping them improve their technical skills and elevate their performance level. Fig 9 presents the average token usage and associated costs across various processes within the AI Table Tennis Coaching System, highlighting its notable cost-effectiveness in several key aspects.

Firstly, the system exhibits high cost-effectiveness in devising match strategies for beginners, with an average cost of only $0.189 per session. This price not only reflects the accessibility of the technology but also ensures broad acceptability, allowing more table tennis beginners to benefit from professional-level guidance.

Secondly, the analysis of technical maneuver errors, an essential part of training, costs $0.134, underscoring the substantial potential of AI technology in reducing training costs. Through precise analysis of technical errors, coaches and athletes can swiftly identify and rectify technical deficiencies, thereby enhancing training efficiency.

Additionally, the cost for error recognition in maneuvers is $0.192, a crucial component for personalized training. The AI system captures and analyzes athletes' movements in real time, promptly identifying issues and providing feedback, which is crucial for improving athlete performance.

Lastly, the long-term training cost amounts to $0.175, reflecting the system's stability and reliability in delivering high-quality training services over an extended period. Long-term training programs guide table tennis beginners through different stages of practice, setting specific goals and focuses for each stage akin to periodized training methods [31], thereby promoting continuous improvement in their technical proficiency. In terms of business models, there is little difference in computation time, resource consumption, and scalability. Subsequently, we will develop a privatized open-source large model for the AI Ping Pong Coach system.

# 4 Discussion

This study utilizes an AI table tennis coaching system that integrates motion capture technology, computer vision trajectory recognition, and a multimodal large language model with a table tennis knowledge base to identify common mistakes made by beginners and provide targeted guidance for these errors. The system has high recognition accuracy, provides comprehensive personalized training suggestions, and has low maintenance and analysis costs.

## 4.1 Identifying common errors in table tennis novices using AI table tennis coaching system

Beginners mastering the fundamentals is crucial in learning table tennis [32]. During table tennis practice, beginners often make mistakes in basic techniques. When a coach is not available to help them correct these mistakes, they can use the AI table tennis coaching system. The AI table tennis coaching system can effectively identify errors in the learners' basic techniques, improving their practice efficiency. Below is the recognition process of the AI table tennis coaching system. The AI table tennis coaching system utilizes advanced large language models, primarily GPT-4, to process and analyze both visual information and text data. Using visual recognition technology, the system tracks the trajectory of the ping-pong ball in real-time with high precision. This includes employing background subtraction techniques to isolate the ball, applying Gaussian blur to reduce image noise, and estimating the ball's position through dynamic differential information. Additionally, by utilizing open-source libraries such as OpenPose, the system can estimate the athlete's posture in real-time [33]. And, The system has built a dedicated table tennis knowledge base by searching for relevant instructional videos, table tennis theory books, and consulting with table tennis coaches and experts. This knowledge base provides the theoretical foundation for the multimodal large language model to identify the learner's technical errors. Furthermore, each type of error is defined by corresponding multimodal input data, such as wrist speed-time graphs, arm speed-time graphs, elbow angle-time graphs, and images of the entire body at the moment of hitting the ball. By integrating these technologies and methods, the AI table tennis coaching system can effectively identify common errors in beginners' table tennis techniques, thereby improving their practice efficiency.

## 4.2 Personalized training recommendations based on comprehensive data analysis

The AI table tennis coaching system conducts in-depth analysis of athletes' performance in both matches and training sessions, not only uncovering the root causes behind each mistake but also providing comprehensive training recommendations on a holistic level. Leveraging advanced data analytics, the system transforms extensive match and training data into intuitive charts and statistics, integrating rich prior knowledge from its repository to tailor personalized training and match strategies for athletes. This approach helps athletes maximize their individual strengths, unlock their potential, and cultivate unique technical styles [34].

From a long-term developmental perspective, by reviewing athletes' competitive histories, identifying trends in technical evolution, psychological maturity, and physical condition changes, the AI coach can pinpoint critical milestones in athletes' growth paths. Based on these analyses, the AI coach not only proposes targeted technical and tactical training plans but also provides long-term developmental guidance and advice from a career planning perspective, such as changes in technical style and fluctuations in competitive form. Simultaneously, the AI coach emphasizes continuous skill enhancement and potential development.

For instance, if an athlete performs poorly against high-speed shots, the AI table tennis coach may suggest training to improve reaction speed and anticipation.

Overall, empowered by a robust knowledge base, the AI table tennis coach not only analyzes errors on a single-game basis but also offers a comprehensive, personalized set of training and match strategy guidelines through in-depth analysis and integrated utilization of athletes' individual data. It focuses not only on immediate performance improvements but also on long-term athlete development and potential exploration, aiming to significantly enhance the training efficiency and competitive performance of novice table tennis players.

## 4.3 Cost analysis of AI table tennis coaches

Research results indicate that AI coaching based on fundamental theories is as effective as human coaching in specific coaching domains [35]. The AI table tennis coaching system, based on multimodal large-scale language models and knowledge bases, provides effective guidance comparable to human coaches. One of its significant advantages is its economic efficiency. By leveraging commercialized multimodal language models, the system effectively controls initial implementation and maintenance costs in technical aspects. The maturity of these models means that developers primarily handle major development and maintenance tasks, substantially reducing economic burdens on users. Regarding hardware requirements, the AI table tennis coaching system maintains low demands. Modern households commonly possess 1080P30 frame cameras sufficient to meet system needs, eliminating the need for additional investment in expensive equipment. Computational and analytical processes can be conducted in the cloud, eliminating the necessity for users to purchase high-performance computing hardware, further lowering the entry barriers. The system's advancement is not only evident in its technological aspects but also in its ease of deployment and excellent cost control capabilities. It is user-friendly for individual users and provides an efficient and economical training aid for sports training institutions and coaches. The current table tennis coaching system utilizes commercial large language model interfaces, with inference conducted in the cloud. However, this approach faces challenges in real-time human motion detection under resource-constrained conditions. Future work could involve deploying quantized open-source multimodal large models, such as Qwen-VL and InternVL, on edge devices to achieve a truly portable table tennis coaching system. With ongoing technological advancements and further cost reductions, AI-based table tennis coaching systems, along with other AI-based sports teaching systems, are expected to achieve broader applications in sports training, driving the entire industry towards smarter and more economical directions. The AI table tennis coaching system, characterized by its low cost and high efficiency, introduces innovative solutions to the table tennis training field, promising a broad prospect for AI technology in sports training, potentially leading to innovations in sports training methodologies.

## 4.4 Adapting the system for semi-professional and professional table tennis players

As the AI table tennis coaching system currently targets beginners, extending its functionalities to cater to semi-professional and professional players represents an opportunity to enhance its impact. For higher-level athletes, more sophisticated training systems are required to accommodate the greater precision, speed, and complexity of their techniques. This adaptation would necessitate advancements in both hardware and software. In terms of hardware, the system might need to incorporate higher frame-rate cameras (e.g., 120fps or higher) for more precise motion capture. These cameras could better analyze the speed and subtle movements of professional-level gameplay, particularly in fast-paced rallies. Additionally, integrating motion

sensors or inertial measurement units (IMUs) on players' bodies could provide richer biomechanical data, enabling the system to assess nuanced footwork, body coordination, and fine motor skills. On the software side, the AI model could be trained on more advanced datasets that capture the unique technical and tactical challenges faced by professional players. For example, high-level tactical patterns, response strategies, and opponent analysis could be incorporated, allowing the system to offer real-time feedback and recommendations tailored to professional athletes' competitive environments. This would expand the system's applicability and provide a training tool that grows with the athlete's skill level. By addressing these aspects, the system can offer more comprehensive, professional-grade training, further advancing its potential as a versatile coaching tool.

Although the significant achievements of the AI table tennis coaching system in identifying beginner errors and providing personalized training guidance, there are also limitations. Firstly, its accuracy in judging footwork errors is relatively low because correct footwork depends not only on shifting body weight but also on the coordination of step size, speed, and stroke action, factors inadequately represented in the current model. Therefore, in order to improve the accuracy of footwork recognition in the AI table tennis coaching system, specific solutions should be implemented in the future, such as integrating a multi-camera system or combining inertial measurement units (IMUs) with visual data. Secondly, the system's performance is influenced by the quality and diversity of the dataset, unable to cover all types of errors and scenarios, leading to biased analysis results. Additionally, the system is primarily aimed at beginners, with limited coaching capabilities for high-level athletes."

## 5 Conclusion

This study constructs an AI table tennis coaching system based on a multimodal large-scale language model integrated with a table tennis knowledge base. The system accurately identifies common mistakes among table tennis beginners and provides targeted training guidance by combining table tennis trajectory recognition, athlete motion capture analysis, and an extensive repository of table tennis expertise. Through deep learning and the analysis of technical data, the system not only significantly enhances training efficiency and quality but also substantially reduces coaching costs. Although the current AI table tennis coaching systems still face challenges such as partial inaccuracies in recognition and a strong dependency on the user's computer proficiency, they nonetheless demonstrate the robust capabilities of multimodal large language models and the potential for popularizing table tennis. Future research will focus on improving the accuracy of footwork recognition and expanding the system's capabilities to support the high-level athletes training. To achieve this, the system will integrate multi-camera setups with inertial measurement units (IMUs) and visual data, enabling more precise tracking of foot movements, body coordination, and weight transfer, which are essential for advanced players. Furthermore, the system will incorporate higher frame-rate cameras (e.g., 120fps or greater) and advanced datasets that capture the tactical complexity and technical precision required in elite-level competitions. These advancements will significantly enhance the system's application in training and contribute to the overall development of table tennis.

## Supporting information

**S1 Appendix. AI table tennis coach system detailed evaluation questionnaire.**
(PDF)

## Author Contributions

**Conceptualization:** Wenlong Ma.

**Data curation:** Wenlong Ma, Wei Xing, Huan Liu.

**Formal analysis:** Wenlong Ma, Wei Xing.

**Investigation:** Wenlong Ma.

**Methodology:** Wenlong Ma, Yang Liu.

**Project administration:** Wenlong Ma.

**Resources:** Wenlong Ma.

**Software:** Wenlong Ma.

**Supervision:** Wenlong Ma, Xutao Liu, Rongji Zhao.

**Validation:** Wenlong Ma, Yang Liu.

**Visualization:** Wenlong Ma, Yang Liu.

**Writing – original draft:** Wenlong Ma.

**Writing – review & editing:** Qing Yi, Rongzhi Li.

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
