## [Decision Letter · Decision Letter 0]

25 Sep 2024

PONE-D-24-31115Table Tennis Coaching System Based on a Multimodal Large Language Model with a Table Tennis Knowledge Base

PLOS ONE

Dear Dr. Ma,

Thank you for submitting your manuscript to PLOS ONE. After careful consideration, we feel that it has merit but does not fully meet PLOS ONE’s publication criteria as it currently stands. Therefore, we invite you to submit a revised version of the manuscript that addresses the points raised during the review process.

The manuscript has been evaluated by three reviewers, and their comments are available below. The reviewers have raised a number of major concerns.

They note several shortcomings in the methodology, including details about inputs to the model and the validity of the evaluation methods. Please take care to significantly revise the Methods, since manuscripts published in PLOS ONE must be reproducible. In addition to the reviewers' concerns, please ensure you have provided sufficient information about the recruitment and demographic information of both the novice players and the experts, and also ensure that all data and code are fully available. If ethics approval was sought please provide details, or else explain why this was not required.

Additional concerns include the limited validation of the tool's functionality as a tool to actually improve table tennis players' abilities  - which renders some of the conclusions overstated - and insufficient discussion of the utility of the tool compared to similar approaches. Limitations of the approach also need to be discussed further.

Could you please carefully revise the manuscript to address all comments raised? Please note that your revisions should be substantial, and the manuscript may be rejected if the concerns are not addressed adequately.

We look forward to receiving your revised manuscript.

Kind regards,

Marianne Clemence

Staff Editor

PLOS ONE

Journal Requirements:

3. You indicated that ethical approval was not necessary for your study. We understand that the framework for ethical oversight requirements for studies of this type may differ depending on the setting and we would appreciate some further clarification regarding your research. Could you please provide further details on why your study is exempt from the need for approval and confirmation from your institutional review board or research ethics committee (e.g., in the form of a letter or email correspondence) that ethics review was not necessary for this study? Please include a copy of the correspondence as an ""Other"" file.

4. Please update your submission to use the PLOS LaTeX template. The template and more information on our requirements for LaTeX submissions can be found at http://journals.plos.org/plosone/s/latex.

“The authors have no conflicts of interest to disclose”

6. In the online submission form, you indicated that [Code is open to the public. Data can be requested via email].

7. Please amend the manuscript submission data (via Edit Submission) to include author Dr. Rongzhi Li.

8. We note that Figures 2,3 and 6 includes an image of a [patient / participant / in the study].

9. We note you have included a table to which you do not refer in the text of your manuscript. Please ensure that you refer to Table 1 in your text; if accepted, production will need this reference to link the reader to the Table.

Reviewers' comments:

Reviewer's Responses to Questions

**Comments to the Author**

1. Is the manuscript technically sound, and do the data support the conclusions?

Reviewer #1: Partly

Reviewer #2: Partly

Reviewer #3: Partly

2. Has the statistical analysis been performed appropriately and rigorously? 

Reviewer #1: Yes

Reviewer #2: No

Reviewer #3: No

3. Have the authors made all data underlying the findings in their manuscript fully available?

Reviewer #1: No

Reviewer #2: No

Reviewer #3: No

4. Is the manuscript presented in an intelligible fashion and written in standard English?

Reviewer #1: Yes

Reviewer #2: No

Reviewer #3: Yes

5. Review Comments to the Author

Reviewer #1: GENERAL CONCEPT COMMENTS

Congratulations to the authors for their diligent work and valuable contribution to the field through this research endeavour. After carefully reviewing the article and considering the recommendations provided, it is evident that the manuscript presents valuable insights into the topic. The article “Table Tennis Coaching System Based on a Multimodal Large Language Model with a Table Tennis Knowledge Base” is an innovative and promising contribution to the intersection of sports and AI. The application of multimodal large language models (LLMs) for table tennis coaching has potential to revolutionize training by providing personalized, precise, and cost-effective guidance. However, there are several areas in which the article can be improved to meet the standards of scientific rigor typically applied in research, particularly in the methods section, which lacks crucial details.

SPECIFIC COMMENTS

1. INTRODUCTION

Specific textual recommendations for accuracy and precision:

(P. 8) - The introduction provides a good foundation on the benefits of table tennis, but the transition to AI application feels abrupt. A stronger link between the issues with current table tennis training methodologies and how AI offers an innovative solution would improve the flow.

• Suggestion: Expand on why current technologies (like VR or traditional video analysis) are insufficient for personalized, high-efficiency training and how multimodal LLMs can bridge this gap.

(P. 8) – Correct 10 and 11 reference formats.

(P. 8) – Correct 12 and 13 reference formats.

(P. 8) - “Children who regularly engage in table tennis training demonstrate better physical composition and fitness, compared to active children who do not participate in regular physical activities[3]” The sentence seems confusing; consider revising it for clarity by specifying the comparison group and the benefits more clearly.

(P. 8) – The acronym “AI” appears before it is introduced in the text. To improve clarity, “Artificial Intelligence (AI)” should be introduced the first time it’s mentioned.

(P. 9) – Correct 20 and 21 reference formats.

(P. 9) – Correct 23 and 24 reference formats.

(P. 9) - The discussion of technological advancements (VR, AI) could benefit from a more structured comparison of existing tools used in sports and how this system differs or improves upon them.

2. METHODS

Consider revising the text to avoid using first-person language, as scientific writing typically favors a more objective tone. For example, instead of “I positioned my smartphone,” use “The smartphone was positioned” to maintain formality and focus on the research process.

Specific textual recommendations for accuracy and precision:

(P. 9) - The method involving the placement of a single smartphone for capturing video is innovative for its simplicity. However, it may limit the system’s ability to capture complex footwork and body dynamics.

• Suggestion: Acknowledge that using only one camera may reduce the accuracy of certain movements (like footwork) and propose future improvements, such as adding multiple camera angles. This limitation should be made clear in the methodology section as well as in the limitations section.

(P. 10) – The figure 1, 2, 3, 4 must appear immediately after its introduction in the text.

(P. 10) - While the system mentions GPT-4 as the core model for providing training recommendations, there is insufficient detail about how the model was trained or adjusted for this specific application. The use of LLMs in a complex task such as table tennis coaching demands more transparency, especially around: (i) Whether the model was pre-trained or adjusted; (ii) The dataset used for fine-tuning (e.g., how much data, and from which sources?); (iii) Hyperparameters such as learning rate, batch size, and optimization algorithms used in training, similar to how traditional statistical methods require reporting of confidence intervals or normality checks; (iv) Was any data augmentation or preprocessing applied to enhance the model’s performance?.

(P. 11) - The integration of motion capture data (from OpenPose) and visual recognition data into the LLM is a significant aspect of the system, but there is no clear explanation of how this data is formatted or fed into GPT-4. The authors must explain how multimodal data is handled, much like a traditional study would report assumptions checks or data transformations before analysis.

Additional questions also rise: (i) Are the visual and motion data converted to text or encoded in another form before being processed by the model?; (ii) If the data is fed into the model as structured inputs, provide details on the data pipeline and preprocessing steps; (iii) Without this information, readers are left uncertain about the complexity of the integration between the multimodal data and the LLM.

(P. 11) - OpenPose is a robust tool for motion capture, but it is important to note that its performance may degrade in cluttered or poorly-lit environments, which could affect amateur users.

• Suggestion: Add a section on the environmental limitations of OpenPose and how it affects data accuracy. Including some recommendations on ideal lighting and camera quality for users could be helpful.

(P. 11) - The core functionality of the system relies on GPT-4 generating recommendations based on visual and motion data inputs. However, there is no mention of the prompts, or the structure of the input data used to query the language model. This is a critical detail for understanding how the model generates athlete-specific advice. The article should provide examples of the exact prompts given to GPT-4, much like one would report the mathematical formulas or procedures used in a statistical test. For instance: (i) What input does the model receive? Is it a natural language description, a set of structured variables, or something else?; (ii) Are there any predefined templates or rules for these prompts?; (iii) This omission weakens the study’s transparency, as it is crucial to understand the interaction between multimodal data and the LLM.

(P. 11) - The study should also address potential concerns about overfitting, particularly because it is difficult for AI models to generalize across varied real-world scenarios without extensive data. It’s important to understand if the system was tested on diverse datasets, akin to how normality or homogeneity tests are used in statistical research. In example: (i) Was the model validated on external datasets or players not included in the original training data? And (ii) Did the authors apply regularization techniques (e.g., dropout layers, early stopping) to prevent overfitting?

(P. 12) - The dataset is well-defined in terms of error categorization (e.g., wrist errors, footwork errors), but the process of how data were annotated by experts could be more detailed.

• Suggestion: Include details on the inter-rater reliability among the coaches who annotated the errors. How consistent were they in identifying these errors? A mention of this would improve the perceived robustness of the dataset.

(P. 12 L. 5–25) - The system’s accuracy in identifying table tennis errors is reported, but crucial details, like confidence intervals or variance in accuracy across different datasets, are missing. Just as traditional statistical analyses require detailed metrics (effect sizes, p-values), AI models must be rigorously evaluated using standard performance metrics. In example: (i) Report accuracy, precision, recall, and F1-scores for error identification; (ii) Include confidence intervals or standard deviations for these metrics to indicate the model’s robustness; (iii) Clarify whether cross-validation was used to ensure that the model is not overfitting to a particular set of data; and (iv) Include a confusion matrix to show where the model succeeds or fails in classifying various errors (e.g., racket position vs. footwork).

(P. 12 L. 10–20) - The article does not clarify whether the outputs generated by GPT-4 were evaluated for quality, accuracy, or consistency. This is essential, as one would expect an evaluation of any AI system’s output similar to how confidence intervals or p-values are reported in statistical studies. As a reviewer I have a few questions: (i) Was the quality of GPT-4’s recommendations assessed? Were there metrics or cross-validation techniques applied to ensure the model consistently provides correct or useful feedback?; (ii) Did the authors perform any error analysis to understand common mistakes made by the model (e.g., false positives or incorrect classifications)?; (iii) If experts evaluated the system, what were the specific criteria used to measure GPT-4’s performance?

(P. 12) – Table 1 should be introduced in the text before the actual table.

3. RESULTS

Specific textual recommendations for accuracy and precision:

(P. 13) – Figure 5 should be introduced in the text prior to its appearance in the document.

(P. 13) – The figure 6 must appear immediately after its introduction in the text.

(P. 13) - The system demonstrates impressive accuracies (73% for arm-related errors and 82% for racket errors). However, these percentages are not sufficient for real-world use if aiming for higher levels of training (beyond beginner-level players).

• Suggestion: Clarify that these accuracy metrics are primarily for beginners and that further refinement is needed for advanced players. Also, suggest ways to increase accuracy, such as incorporating biomechanics data or higher-resolution tracking.

(P. 14) - The comparison between GPT-4, Gemini Pro, and Qwen-VL is useful, but the decision criteria for selecting GPT-4 should be elaborated.

• Suggestion: Add more detailed metrics to the model comparison (e.g., computation time, resource consumption, scalability) and explain why GPT-4 was chosen over the others in those dimensions, not just for accuracy and false positives.

(P. 14) – Figure 7 should be introduced in the text prior to its appearance in the document.

(P. 15) – Figure 9 should be introduced in the text prior to its appearance in the document.

4. DISCUSSION

Specific textual recommendations for accuracy and precision:

(P. 16) - The cost analysis focuses primarily on beginner-level applications, but there is no mention of scaling this system for more advanced or professional users.

• Suggestion: Include a section that discusses how the system could be adapted for semi-professional or professional players, with potentially more advanced (and more expensive) hardware. This would broaden the scope of the application and make the study more impactful.

(P. 16) - The lower accuracy in recognizing footwork errors is mentioned, but more detail should be given regarding how this will be improved in future research.

• Suggestion: Propose specific technical solutions, such as integrating a multi-camera system or combining inertial measurement units (IMUs) with visual data. Without a clear plan, it may appear that this limitation will remain unaddressed.

5. CONCLUSION

Specific textual recommendations for accuracy and precision:

(Page 17) - The conclusion touches on future improvements in footwork recognition and training for high-level athletes, but these should be expanded with more concrete plans.

6. REFERENCES

In most of the references, the symbol ‘[J]’ appears. Please confirm whether this is intentional.

Reviewer #2: Summary: This work introduces an AI-driven TT coaching system using a Multimodal LLM with a table tennis knowledge base, aiming to provide precise training guidance and match strategies for TT novice players eventually improving the overall performance.

Merits:

The paper tackles an interesting research domain.

Demerits and Questions:

There is a gap between highlighting the novel contributions as the approaches have been utilized for the study. Can the authors explicitly discuss and highlight the novelty of the paper? As the authors employed existing approaches.

Please conduct analysis and provide the ANOVA/Cohen's kappa scores based on the experts rating.

The authors envision employing the proposed TT coaching system will enhance training efficiency, if so can the authors explicitly show the improvement rate before and after the training session?

We encourage the authors to release the data and code during the review process.

There is a lack of related literature comparison as to where this work fits and outperforms SOTA methods? - Please provide a tabular fashion comparison?

Please provide an ablation study or rationale for adopting the proposed pipeline? We found this part of the motivation is lacking throughout the paper?

Discuss the proposed framework's limitations, particularly scalability and real-world deployment, to provide a more balanced perspective on its applicability.

Please provide the feasibility test on how the authors plan to deploy in resource-constrained devices.

Reviewer #3: Minor formatting issues:

1.1 Inconsistent citation format: "physical condition[12,[13]" and "unaffected by external conditions[10,[11]". Please standardize the citation format throughout the paper.

1.2 A reference for OpenPose is needed.

1.3 Figure formatting issues in the main content: In sections 2.1 and 2.2, only figure captions are present without the actual figures.

1.4 “In Figure 6, it can be seen that in an 11-point

match analyzed by the AI table tennis coaching system, a total of 41 points were lost. Among these,

there were 9 errors due to arm movements and racket angle, 7 errors due to hitting position and

slow body transitions, 3 errors due to mistiming of strokes, 1 error due to lagging footwork, 2

errors due to wrist movements, and 3 errors due to high center of gravity.” This sentence should be changed to improve clarity. For example, it should clarify whether arm movements and racket angle errors appeared 9 times each or 9 times in total during the match

2 The authors state that data and code are publicly available without restrictions. However, no direct links to the code and data are provided. Including an anonymous GitHub repository would provide more details about the data and code framework for reproduction purposes and help people to understand the framework.

3 The authors mention a knowledge base that provides the theoretical foundation for MLLMs. It would be beneficial to clarify whether this knowledge base is text-based or video-based. A more detailed explanation or an example would help illustrate this clearly.

4 It would be valuable to compare the performance of state-of-the-art MLLMs on this task. A comparison between the authors' model and SOTA MLLMs would help clarify the contribution of this work.

The paper lacks a clear explanation of how visual recognition, motion capture, and large language models are integrated. How is the video data incorporated into the MLLM models? The authors use pre-trained recognition algorithms to identify players and ball trajectories, but the advantages of this approach are not clearly stated. Does this perform better than directly uploading the video to the MLLMs? A detailed results comparison would help support the authors' claims.

5 In Table 1, the meaning of "Proportion" and "Quantity" is unclear. Does this represent the test subset or the full dataset proposed by the authors? The proportions do not sum to 100%. A better clarification of the dataset would help readers understand the size of the dataset and the proportion used as the test set.

6 In Figure 7, the authors claim "Automated loss analysis, error cause investigation, and recommendations for a table tennis novice in an 11-point match." The AI-generated suggestions seem applicable to most people who make such errors. There is a concern that the generated suggestion content is based on error type rather than personalized movement videos. Is it possible to prompt the model to show different arm angles where the player makes mistakes, since Figure 3 suggests arm angles can be captured? This would make the model's output appear more personalized.

6. PLOS authors have the option to publish the peer review history of their article (what does this mean?). If published, this will include your full peer review and any attached files.

Reviewer #1: **Yes: **Tatiana Peixoto Sampaio

Reviewer #2: No

Reviewer #3: No

---

## [Author Response · Author response to Decision Letter 0]

27 Nov 2024

Dear Editors of PLOS ONE,

Greetings!

We sincerely thank you for your thorough review of our manuscript and for providing detailed publication recommendations. We have carefully addressed each of the points raised and have made the necessary revisions to our submission. Below, we outline our responses to each requirement:

1.Manuscript Style and File Naming

We have ensured that our manuscript adheres to PLOS ONE's style requirements, including the correct file naming conventions. All sections of the manuscript have been adjusted accordingly to meet these guidelines.

2.Code Sharing Compliance

The de-identified raw data used in this study will be made available in a public online repository following publication, Preprocessing and analysis code has been made available in a public GitHub repository:

https://github.com/mwlsus/ttcs_by_llm

This ensures that our code follows best practices and facilitates reproducibility and reuse of our research.

3.Ethical Approval Clarification

Ethical approval

Dear Editor：

The focus of this research is on the development of an AI-based table tennis coaching system. This system primarily involves the use of motion capture, computer vision, and AI-based data analysis to improve the accuracy and effectiveness of table tennis training. The study employs visual recognition technology to analyze ball trajectories and identify common technical errors among beginners.

 Specifically:

1.Non-identifiable Data Collection: Data are anonymized and involve only technical and movement data (e.g., ball trajectory, racket positioning, and limb motions) captured through visual recognition technologies. No sensitive or personal identifiers are involved.

2.Minimal Risk: The study's focus on low-risk data collection and the use of standard sports monitoring equipment (such as a 1080p camera on a tripod) reduces the possibility of risk to participants. Additionally, the captured data solely include general movements rather than individual-specific biometric informationPublic Settings and Full Consent: Participants were briefed about the study's purpose and scope, with informed consent obtained before data collection. Data protection measures, such as encryption and anonymization, further support privacy compliance.

3.A total of 10 table tennis experts were selected to evaluate thepotential of the AI-based table tennis coaching system. The survey was primarily distributed through online platforms such as WeChat and Wenjuanxing. All participants were adults (18 years or older). Participants filled out the questionnaire by clicking through it continuously. Before completing the questionnaire, they were informed of the purpose of the survey and were notified that submitting the questionnaire would be considered as informed consent. Participants could exit the questionnaire at any time during the process.

Given the nature of this data (non-sensitive, anonymized, and low-risk) and the fact that it does not involve intrusive data collection, this study aligns with the institution's policy of not requiring formal ethical review. The research emphasizes technical advancements in AI coaching systems rather than personal data analysis, thus maintaining ethical standards without needing specific ethics committee approval.Because my experiment started at Beijing Sport University, I have obtained the stamp from the Ethics Review Committee of Beijing Sport University.

 Yours sincerely，

 Wenlong Ma

 November 8, 2024

4.Competing Interests Statement

We have updated the Competing Interests section in the online submission form to state: "The authors have declared that no competing interests exist.". This statement is also included in our cover letter for your reference.

5.Data Availability

Following PLOS ONE's data availability policy, all data underpinning our findings have been deposited in the Github repository:

https://github.com/mwlsus/ttcs_by_llm

We have updated the manuscript to include these links. Additionally, we have ensured that no data breaches ethical or legal standards, and any sensitive information has been appropriately anonymized.

6.Author Information Addition

 Wenlong Ma1,（The first author）

Yang Liu2,

Qing Yi3,

Xutao Liu4,

Wei Xing5,

Rongji Zhao 6 ,

Huan Liu 7 ,

 Rongzhi Li1*（Corresponding author）

7.Consent for Publication of Images

For Figures 2 and 3，I have already included the revision in the introduction to the methods section manuscript."The revision is as follows：

For Figures 2 and 3,the individual in this research has given written informed consent (as outlined in PLOS consent form) to publish these case detail.

For Figure6 I have already included the revision in the 3.2 Guidance for Unforced Errors in Table Tennis Novices in the manuscript The revision is as follows：

For Figure6,the individual in this research has given written informed consent (as outlined in PLOS consent form) to publish these case detail.

8.Table Reference in Manuscript

We have reviewed the manuscript and ensured that Table 1 is appropriately referenced within the text. Relevant sections now include explicit mentions of Table 1 to guide readers effectively.

Once again, we appreciate the constructive feedback and guidance provided by the PLOS ONE editorial team. We believe that these revisions have strengthened our manuscript, and we look forward to your favorable consideration.

Response to Reviewer1

Response to GENERAL CONCEPT COMMENTS：Congratulations to the authors for their diligent work and valuable contribution to the field through this research endeavour. After carefully reviewing the article and considering the recommendations provided, it is evident that the manuscript presents valuable insights into the topic. The article “Table Tennis Coaching System Based on a Multimodal Large Language Model with a Table Tennis Knowledge Base” is an innovative and promising contribution to the intersection of sports and AI. The application of multimodal large language models (LLMs) for table tennis coaching has potential to revolutionize training by providing personalized, precise, and cost-effective guidance. However, there are several areas in which the article can be improved to meet the standards of scientific rigor typically applied in research, particularly in the methods section, which lacks crucial details.

Response：Thank you for your insightful and constructive feedback on our article. We truly appreciate your recognition of our efforts and the potential impact of our research on AI-driven table tennis coaching systems. Your comments regarding the innovative nature of applying multimodal large language models in this domain are very encouraging.We acknowledge the need for more detail in the methods section and will carefully address this to enhance clarity and replicability. Your feedback highlights key areas for improvement, and we are committed to refining our manuscript to meet the highest standards of scientific rigor. Thank you once again for your valuable guidance.

1.Introduction：

Response to comments: 1.Specific textual recommendations for accuracy and precision:(P. 8) - The introduction provides a good foundation on the benefits of table tennis, but the transition to AI application feels abrupt. A stronger link between the issues with current table tennis training methodologies and how AI offers an innovative solution would improve the flow.Suggestion: Expand on why current technologies (like VR or traditional video analysis) are insufficient for personalized, high-efficiency training and how multimodal LLMs can bridge this gap.

Response：I introduced the numerous benefits of table tennis. Following this, I wrote,Interest in learning table tennis can inspire individuals to be more willing to participate in the sport, thereby demonstrating the numerous benefits of playing table tennis，and point out an effective and scientifically-based table tennis training method can stimulate table tennis learners' interest.However, the current table tennis training methods are relatively limited, still adhering to a teacher-centered approach where instructors lecture while students listen. Additionally, teachers predominantly follow traditional textbook-based teaching methods, which lack personalized instruction and place students in a passive learning state, inevitably affecting their interest in learning table tennis skills.The evolution of technology has markedly advanced table tennis training methods, integrating various technological tools to augment training efficiency and outcomes[6]-7]. The application of Virtual Reality (VR) technology enables learners to train within simulated environments[8], which not only heightens learner engagement but also allows for specialized skill practice unaffected by external conditions[9]-10].However, VR technology exhibits significant limitations in delivering precise and personalized guidance. VR training environments primarily emphasize scene recreation and often lack the capability to dynamically adjust based on real-time feedback from learners. As a result, VR are unable to provide targeted recommendations for technical improvement. While the benefits of VR technology have been introduced earlier, here, we turn to highlight its limitations, specifically its inability to provide personalized training suggestions and plans, and why this occurs. This then leads to the connection with AI applications and introduces what AI is.In recent years，AI has made significant advancements，it can provide personalized training recommendations and plans for beginners, strengthening their understanding of technical movement learning. Moreover, it lays a solid foundation for beginners to enhance their proficiency in technical skills.Artificial intelligence is the science of endowing programs with the ability to change themselves for the better as a result of their own experiences[11],which sets the stage for the introduction of multimodal large language models in the following section.

Response to comments: 2.Children who regularly engage in table tennis training demonstrate better physical composition and fitness, compared to active children who do not participate in regular physical activities[3]” The sentence seems confusing; consider revising it for clarity by specifying the comparison group and the benefits more clearly.

Response：Table tennis players show better cardiovascular endurance, muscle strength, and flexibility compared to physically peers not engaged in regular sports training[3].

Response to comments: 3.The acronym “AI” appears before it is introduced in the text. To improve clarity, “Artificial Intelligence (AI)” should be introduced the first time it’s mentioned.

Response：Artificial intelligence is the science of endowing programs with the ability to change themselves for the better as a result of their own experiences[11].

Response to comments: 4.The discussion of technological advancements (VR, AI) could benefit from a more structured comparison of existing tools used in sports and how this system differs or improves upon them.

Response：Comparison of technological advances: Thank you for the suggestion. We will structure the comparison of current sports tools more clearly and further explain the unique aspects and improvements of the proposed system. In the text, the differences of the AI Table Tennis Coaching System have already been described in detail.This system effectively identifies common errors among beginners and offers targeted guidance and training recommendations. By compiling and analyzing extensive data on table tennis techniques, this study has constructed a comprehensive dataset that supports further research and enhances the AI system’s capabilities. AI table tennis coaching system significantly improves training efficiency and quality while substantially reducing coaching costs.

2. METHODS

Response to comments: 1.Consider revising the text to avoid using first-person language, as scientific writing typically favors a more objective tone. For example, instead of “I positioned my smartphone,” use “The smartphone was positioned” to maintain formality and focus on the research process.

Response：Thank you for your feedback! I appreciate the suggestion to adopt a more objective tone. I’ll revise the text to eliminate first-person language, ensuring the research is presented with clarity and formality.A smartphone was positioned on a tripod to capture clear angles of beginners' forehand, backhand, and serving techniques.

Response to comments: 2.(P. 9) - The method involving the placement of a single smartphone for capturing video is innovative for its simplicity. However, it may limit the system’s ability to capture complex footwork and body dynamics.• Suggestion: Acknowledge that using only one camera may reduce the accuracy of certain movements (like footwork) and propose future improvements, such as adding multiple camera angles. This limitation should be made clear in the methodology section as well as in the limitations section.

Response：Thank you for your thoughtful insights.I added sentences about using a single camera for motion capture in the introduction section of the methodology."While using a single camera offers simplicity, it may reduce the accuracy in capturing certain movements, particularly footwork. Future improvements could involve the addition of multiple camera angles to enhance precision in detecting footwork and other complex body dynamics.

Response to comments: 3.(P. 10) – The figure 1, 2, 3, 4 must appear immediately after its introduction in the text.

Response：Thank you for your comment! I will ensure that Figures 1, 2, 3, and 4 are placed immediately after their respective introductions in the text for better clarity and flow.

Response to comments: 4.(P. 10) - While the system mentions GPT-4 as the core model for providing training recommendations, there is insufficient detail about how the model was trained or adjusted for this specific application. The use of LLMs in a complex task such as table tennis coaching demands more transparency, especially around: (i) Whether the model was pre-trained or adjusted; (ii) The dataset used for fine-tuning (e.g., how much data, and from which sources?); (iii) Hyperparameters such as learning rate, batch size, and optimization algorithms used in training, similar to how traditional statistical methods require reporting of confidence intervals or normality checks; (iv) Was any data augmentation or preprocessing applied to enhance the model’s performance?.

Response: Thank you for your valuable feedback, which has helped us improve our manuscript. Our model is based on GPT-4V and achieves recognition and analysis of unforced errors in table tennis through the collaboration of multimodal input, prompt engineering, and a knowledge base. Since the GPT model has already acquired knowledge about human motion and sports theory during training, our table tennis coach model can effectively guide beginners by activating and integrating the pre-trained prior knowledge of the GPT model using multimodal input, prompt engineering, and a knowledge base. To demonstrate the superiority of this pipeline, we have added a comparative analysis in Section 3.1, illustrating the prediction accuracy of table tennis errors by GPT without a knowledge base and preprocessing, compared to our AI table tennis coach system.

Response to comments: 5.(P. 11) - The integration of motion capture data (from OpenPose) and visual recognition data into the LLM is a significant aspect of the system, but there is no clear explanation of how this data is formatted or fed into GPT-4. The authors must explain how multimodal data is handled, much like a traditional study would report assumptions checks or data transformations before analysis.Additional questions also rise: (i) Are the visual and motion data converted to text or encoded in another form before being pro

---

## [Decision Letter · Decision Letter 1]

7 Jan 2025

Table Tennis Coaching System Based on a Multimodal Large Language Model with a Table Tennis Knowledge Base

PONE-D-24-31115R1

Dear Dr. Li,

We’re pleased to inform you that your manuscript has been judged scientifically suitable for publication and will be formally accepted for publication once it meets all outstanding technical requirements.

Kind regards,

Peter Andreas Federolf

Academic Editor

PLOS ONE

Additional Editor Comments (optional):

Reviewers' comments:

Reviewer's Responses to Questions

**Comments to the Author**

1. If the authors have adequately addressed your comments raised in a previous round of review and you feel that this manuscript is now acceptable for publication, you may indicate that here to bypass the “Comments to the Author” section, enter your conflict of interest statement in the “Confidential to Editor” section, and submit your "Accept" recommendation.

Reviewer #2: All comments have been addressed

2. Is the manuscript technically sound, and do the data support the conclusions?

Reviewer #2: Yes

3. Has the statistical analysis been performed appropriately and rigorously? 

Reviewer #2: Yes

4. Have the authors made all data underlying the findings in their manuscript fully available?

Reviewer #2: Yes

5. Is the manuscript presented in an intelligible fashion and written in standard English?

Reviewer #2: Yes

6. Review Comments to the Author

Reviewer #2: All my comments are adequately addressed. Thank you so much to the authors for their hard work and and for releasing the codes for the research community .

7. PLOS authors have the option to publish the peer review history of their article (what does this mean?). If published, this will include your full peer review and any attached files.

Reviewer #2: **Yes: **Indrajeet Ghosh

---

## [Editor Report · Acceptance letter]

16 Jan 2025

PONE-D-24-31115R1 

PLOS ONE

Dear Dr. Li, 

I'm pleased to inform you that your manuscript has been deemed suitable for publication in PLOS ONE. Congratulations! Your manuscript is now being handed over to our production team.

Kind regards, 

on behalf of

Dr. Peter Andreas Federolf 

Academic Editor

PLOS ONE